# Inverting Rayleigh surface wave velocities for crustal thickness in eastern Tibet and the western Yangtze craton based on deep learning neural networks

Xianqiong Cheng[1], Qihe Liu[2], Pingping Li[1], Yuan Liu[1]

[1] College of Geophysics, Chengdu University of Technology, Chengdu , P.R. China
[2] School of Information and Software Engineering, University of Electronic Science and Technology of China, Chengdu , P.R. China

*Correspondence to:* Xian-Qiong Cheng(chxq@cdut.edu.cn)

**Abstract.** Crustal thickness is an important factor affecting lithospheric structure and deep geodynamics. In this paper, a deep learning neural network based on stacked sparse auto-encoder is proposed for the inversion of crustal thickness in eastern Tibet and Western Yangtze craton. First, with the phase velocity of Rayleigh surface wave as input and the theoretical crustal thickness as output, 12 deep-sSAE neural networks are constructed, which are trained by 380,000 and tested by 120,000 theoretical models. We then invert the observed phase velocities through these twelve neural networks. According to the test error and misfit of other crustal thickness models, the optimal crustal thickness model is selected as the crustal thickness of the study area. Compared with other ways detected crustal thickness such as seismic wave reflection and receiver function, we adopt a new way for inversion of earth model parameters, and realize that deep learning neural network based on data driven with the highly nonlinear mapping ability can be widely used by geophysicists, and our result has good agreement with high-resolution crustal thickness models. Compared with other methods, our experimental results based on deep learning neural network and new Rayleigh wave phase velocity model reveal some details: there is a northward-dipping moho gradient zone in Qiangtang block, and relatively shallow northwest-southeast orientation crust at Songpan-Ganzi block. Crustal thickness around Xi'an and Ordos basin is shallow about 35km. Change of crustal thickness in Sichuan-Yunnan block is sharp, where crustal thickness is 60km in northwest and 35km in southeast. We conclude that deep learning neural network is a promising, efficient and believable geophysical inversion tool.

**Keywords:** Crustal thickness; Phase velocities; Surface wave; Stacked sparse auto-encoder; Deep learning ; Neural network

## 1 Introduction

The Eastern Tibet and the western Yangtze craton are one of the key areas for understanding the collision process between the Indo-European plate, and an important area for understanding the collision and contact relationship between the Qinghai-Tibet Plateau and the Yangtze craton. In the field of geosciences, because of the strong seismic activity, the nature of the two blocks is different, especially the special topography. The altitude of the two blocks suddenly rises from about 500 meters in eastern Tibet to 5000 meters in Western Yangtze craton. Many researches focus on understanding the crust and upper mantle structure in this region, especially there have been heated debates on crustal thickness. The discontinuity between the crust and the mantle is called Moho discontinuity, which varies greatly on a small scale and is an important factor in geodynamics, including crustal evolution, tectonic activity, gravity correction of crustal effect, seismic tomography and geothermal models. Many studies focus on obtaining the depth of moho discontinuity called crustal thickness by various data and different methods.

Usually, the crustal thickness can be inverted by many kinds of data, such as inversion of deep seismic sounding profiles in China mainland for crustal thickness (Zeng et al., 1995), inversion of satellite gravity data for global crustal and lithospheric thickness (Fang et al., 1999), inversion of Bouguer gravity and topography data to calculate the crustal thickness of China and its surrounding areas (Huang et al., 2006; Guo et al., 2012), inversion receiver function is used to calculate the crustal thickness and Poisson's ratio of China mainland(Chen et al.,2010;Zhu et al.,2012;Xu et al.,2007). Especially,  one of the newest models crust1.0 at $1^o \times 1^o$(Laske et al.,2013;Stolket al., 2013) is based on refraction and reflection seismology as well as receiver function studies. Besides these data related to crustal thickness mentioned above, crust thickness has significant effects on fundamental mode surface waves (Meier et al.,2007,Grad et al.,2009). Dispersion characteristic of surface wave provides a powerful tool to research structure of

crust and upper mantle (Legendre, C. P. et al.,2015). So far phase and group velocity measurements of fundamental mode surface waves are most commonly used to constrain shear-velocity structure in the crust and upper mantle on a global scale (Zhou et al. 2006;Shapiro &Ritzwoller ,2002) or on regional scale (Zhang et al.,2011;Yi et al.,2008), also the newly developed ambient noise surface wave tomography has been used to constrain shear-velocity structure(Sun et al.,2010;Yaoet al.,2006;Zheng et al.,2008;Zhou et al.,2012),while a few works to invert fundamental mode surface wave data for global or regional crustal thickness and to present a global or regional crustal thickness model(Devile et al.,1999;Meier et al.,2007; Das &Nolet 2001; Lebedev et al.,2013 ). Although the measurement period and method of group velocity and phase velocity are different, also the detection depth and measurement error are different, phase velocity is more sensitive to deeper structure, so it is easier to infer deep structure from phase velocity measurements. We use phase velocity as input to infer the crustal thickness.

There are several inversion methods to get crustal thickness which can be broadly classified into two classes: (1) model-driven method and (2) data-driven method. For the model-driven method, the researchers mainly consider the physical relation between earth parameters space and data space to calculate inversion function. Most model-driven methods deal with the inversion of crustal thickness as a linear problem. More importantly, their results largely depend on the initial earth model. Compared with model-driven method, another fully non-linear data-driven method is called neural network, which is used to obtain crustal thickness (Devile et al.,1999; Meier et al.,2007). Data-driven, highly non-linear mapping neural networks are widely used in geophysical inversion methods, which use actual seismic, logging data and their attributes to predict the earth's parameters. Compared with model-driven inversion, data-driven inversion does not need to consider the physical relationship between the parameters of the earth model and data space, and can map and predict arbitrary non-linear relationship quickly and accurately. Neural networks can be very useful in situations where the forward relation is known, but the inverse mapping is unknown or difficult to establish by more conventional analytical or numerical methods(de Wit et al.,2013). So the target of neural network inversion is to find the mapping from a set of training data. Neural networks have been widely used in different geophysical applications well summarized by van der Baan &Jutten (2000) such as in electrical impedance tomography(Lampinen, J. &Vehtari, A ,2001), in seismic processing including trace editing, travel time picking, horizon tracking, and velocity analysis. Devilee et al.(1999) were the first to use a neural network to invert surface wave velocities for Eurasian crustal thickness in a fully non-linear and probabilistic manner. Meier et al.(2007) further develop the methods of Devilee et al. (1999), then invert surface wave data for global crustal thickness on a $2\circ \times 2\circ$ grid globally using a neural network.

As seismology points out that there are many factors affect phase velocity, inverting phase velocity for discontinuities within the earth forms a non-linear inverse problem (Meier et al.,2007). Because of strong non-linear relations between crust thickness and surface wave dispersion, we cannot treat it with a linear inverse problem as Montagner&Jobert (1988) stated. Although shallow neural network with less number of hidden layers, can present nonlinear inverse function, it maybe cannot learn or approximate the true inverse function well when the true inverse function is too complicated. In contrast, deep learning neural network can overcome this defect since it has powerful representation abilities and can discover intricate structures in large data sets, because it take use of the back-propagation algorithm to indicate how a machine should change its internal parameters that are used to compute the representation in each layer from the representation in the previous layer (LeCunet.al.,2015).

In this paper, in view of the advantages and characteristics of deep learning neural network, a new fast inverse method based on data-driven, called deep stacked Sparse Auto-encoders (sSAE) neural network is introduced to solve the nonlinear geophysical inverse problems. We focus on deep learning neural networks to solve the non-linear inverse problem, and then apply them to retrieve the crustal thickness for eastern Tibet and western Yangtze craton from newest and high-resolution phase velocity maps. Based on normal mode theory we compute phase velocities for the sampled radially symmetric earth models to generate 500,000 theoretical models. First, the theoretical phase velocity of Rayleigh surface wave under random noise is used as input to enhance the robustness of the neural network, and the corresponding theoretical crustal thickness is used as output. Twelve deep neural networks are constructed trained by 380,000  and tested by 120,000 synthetic models. We then invert the observed phase velocities through these twelve neural networks.  According to test errors and misfits with other crustal thickness models, the optimal crustal thickness model is selected as the crustal thickness of the study area.

To the best of our knowledge, we are the first to introduce deep learning neural networks to learn and invert crustal thickness, and our results show that crustal thickness is strong nonlinear with respect to phase velocity. The merits of our methods include:  First, since deep learning neural networks can represent complex functions, it is possible to learn the crustal thickness inverse function precisely. Using

deep learning neural network, we can learn the relationship between surface wave phase velocity and model parameters on the basis of large data (i.e. 500,000 theoretical models in this study), and relax the priori constraints on model parameters (the crustal thickness is limited to 20-100 km).Secondly, inverse mapping based on neural network is of high efficiency because new observations can be inverted instantaneously once well-trained deep learning neural networks with multiple hidden layers are constructed. Thirdly, we can invert any combination of model parameters without resampling model space (we will invert crustal thickness and shear wave velocity simultaneously in future work). Last but not least, the results show that when the number of hidden layers reaches 6 and the test error is about 4.5e-6, the change of the number of neurons in each layer has little effect on the test error, which indicates that the deep learning neural network has strong robustness to the neural network structure with appropriate layers. In the following, we will first briefly introduce the deep learning neural network .

## 2 Deep Learning Neural Networks

In geophysics the real inverse function is usually a very complicated one between data space and model space. The traditional linear inverse methods treat the real inverse function as linear one can resolve the linear relation problems. However, they depend on physical relationships between the parameter space and initial earth model. Neural network has its origins in attempts to find mathematical representations of information processing in biological systems (Bishop ,1995).  The deeper strength of Artificial Neural Networks (ANNs) is, the more capabilities learn to infer complex, non-linear, underlying relationships without any a priori knowledge of the model(Bengio,2009). Shallow neural network has gained in popularity in geophysics last decade and has been applied successfully to a variety of problems such as well-log, interpretation of seismic data, geophysical inversion, etc. Although shallow neural network can present nonlinear inverse function, it can only learn the relatively simple inverse function. In contrast, Many research results indicate that deep learning neural network has powerful representation ability and can apply a big geophysical observable data to learn and approximate the complicated inverse function well( Lecun et al.,2015 Bengio et al.,2006; Liu et al.,2015).

Based on the analysis above, we design deep learning neural network to obtain crustal thickness for eastern Tibet and western Yangtze craton. Compared with shallow neural networks, deep learning neural network allows computational models that are composed of multiple processing layers to learn representations of data with multiple levels of abstraction and can learn complex functions.The essence of deep learning is building an artificial neural network with deep structures to simulate the analysis and interpretation process of human brain for data such as image, speech, text, etc. However, many research results suggest that gradient-based training of a deep neural network gets stuck in apparent local minima, which leads to poor results in practice (Bengio, 2009). Fortunately, the greedy layer-wise training algorithm proposed by Hinton et.al 2006 overcomes the optimization difficulty of deep networks effectively. The training processing of deep neural networks is divided into two steps. First, unsupervised learning methods are employed to pre-train each layer parameters with the output of the previous layer as input, giving rise to initialize parameter values. After that, the gradient-based method is used to finely tune the whole neural network parameter values with respect to a supervised learning criterion as usual. The advantage of the unsupervised pre-training method at each layer can help guide the parameters of that layer towards better regions in parameter space(Bengio,2009).There are multiple types of deep learning neural network, such as convolutional neural networks, deep belief net and stacked Sparse Auto-encoders(sSAE). sSAE works very well in learning useful high-level feature for better representation of input raw data. Since sSAE learning algorithm can automatically learn even better feature representations than the hand-engineered ones, sSAE is used widely in many domains such as computer vision, audio processing, and natural language processing[Hinton,2006; Deng,J et al.,2013]. Similar to these problems, we need extract earth feature representation from dispersion of surface wave. Here we introduce Sparse Auto-encoder briefly, and detailed description of the network training method is given by Liu et al.( Liu et al ,2015).

The structure of sSAE is stacked by sparse auto-encoders to extract abstract features. A typical Sparse Auto-Encoder (SAE) can be seen as a neural network with three layers, as shown in Figure 1, including one input layer, one hidden layer, and one output layer. The input vector and the output vector are denoted by $v$ and $\hat{v}$, respectively. The matrix W is associated with the connection between the input layer and the hidden layer. Similarly, the matrix $\widehat{W}$ connects the hidden layer to the output layer. The vector $b$ and $\hat{b}$ are the bias vectors associated with the units in the hidden layer and the output layer, respectively. The SAE is trained to encode the input vector $v$ into some representation so that the input can be reconstructed from that representation. Let $f(x)$ denote the activation function, and the activation vector of the hidden layer then is calculated (with an encoder) as:

$$z=f(Wv+b), \qquad (1)$$

where $z$ is the encoding result and some representation for the input $v$. The representation z, or code is then mapped back (with a decoder) into a construction $\hat{v}$ of the same shape as $v$. The mapping happens through a similar transformation, e.g.:

$$\hat{v} = f(\widehat{W}z + \hat{b}) \qquad (2)$$

SAE is an unsupervised learning algorithm which sets the target values to be equal to the inputs and constrain output of hidden layer which are near to zero and most hidden layer are inactive, the cost function is expressed as:

$$J_{sparse}(W,b) = J(W,b) + \beta \sum_{j=1}^{S_2} \rho \log\frac{\rho}{\hat{\rho}_j} + (1-\rho)\log\frac{1-\rho}{1-\hat{\rho}_j} \qquad (3)$$

Here $J(W,b)$ is cost function without sparsity constraint, $\beta$ controls the weight of the sparsity penalty term, $S_2$ is the number of neurons in the hidden layer, and the index j is summing over the hidden units in our network. $\hat{\rho}_j$ is the average activation of hidden unit j, $\rho$ is a sparsity parameter, typically a small value close to zero.

Further, a stacked Sparse Auto-Encoder (sSAE) is a neural network consisting of multiple layers of SAE in which SAE are stacked to form a deep neural network by feeding the representation of the SAE found on the layer below as input to the current layer. Using unsupervised pre-training methods, each layer is trained as sSAE by minimizing the error in reconstructing its input which is the output code of the previous layer. After all layers are pre-trained, we add a logistic regression layer on top of the network, and then train the entire network by minimizing prediction error as we would train a traditional neural network. For example, a sSAE with two hidden layers is shown in Figure 2. This sSAE is composed of two SAEs. The first SAE consists of the input layer and the first hidden layer, and the representation or code of the input v is $h_1 = f(W_1 v + b_1)$. The second SAE comprises of two hidden layers, and the code of $h_1$ is $h_2 = f(W_2 h_1 + b_2)$. Each SAE is added to a decoder layer as shown in Figure 1, and we can then employ unsupervised pre-training methods to train each SAE by expression (1). Finally, the matrix $W_1, W_2$, bias vector $b_1$ and $b_1$ are initialized. We then apply supervised fine-tuning methods to train entire network. Since our aim is calculating crustal thickness and this is a regression problem, we attach a layer connected fully with last layer of the encoder part (the matrix $W_s$). After that, we train this network as done in a traditional neural network.

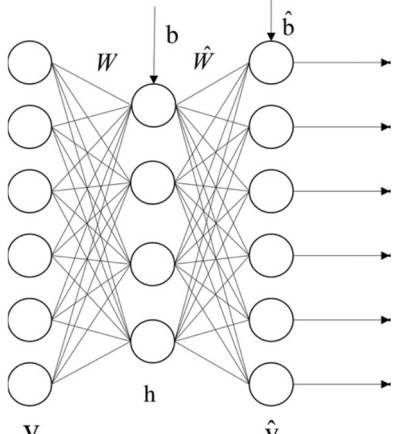

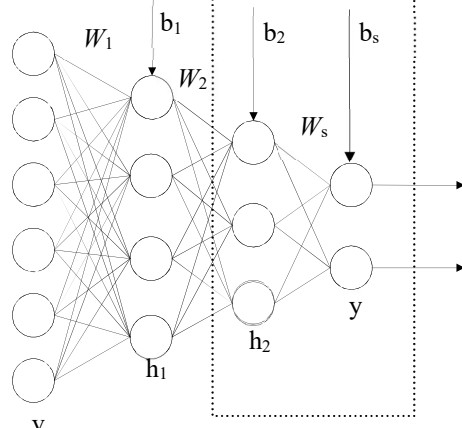

**Figure 1.** An auto-encoder with one hidden layer.(Liu etal.,2015)

**Figure 2.** Stacked Sparse Auto-Encoder with two hidden layers

### 3   Inverting surface wave data for crustal thickness

As Meier et al. (2007) demonstrated that the neural network approach for solving inverse problems is best summarized by three major steps as shown in Figure 3: (1) forward problem. In this stage we proceed by randomly sampling the model space and solve the forward problem for all visited models based on seismic wave normal mode theory. (2) designing a neural network structure. In this stage taking phase velocities with random noise as input and theoretical crustal thickness as output we train the deep learning neural networks and the optimized neural network is obtained. (3) inverse problem. Base on trained networks we invert crustal thickness from observed phase velocities.

In what follows we introduce how to train a sSAE deep learning neural networks to model surface wave dispersion based on synthetic seismogram, and then invert dispersion curves based on the trained

networks. Finally we compare our crustal model with other crustal thickness models, and discuss the geodynamic implications of our model .

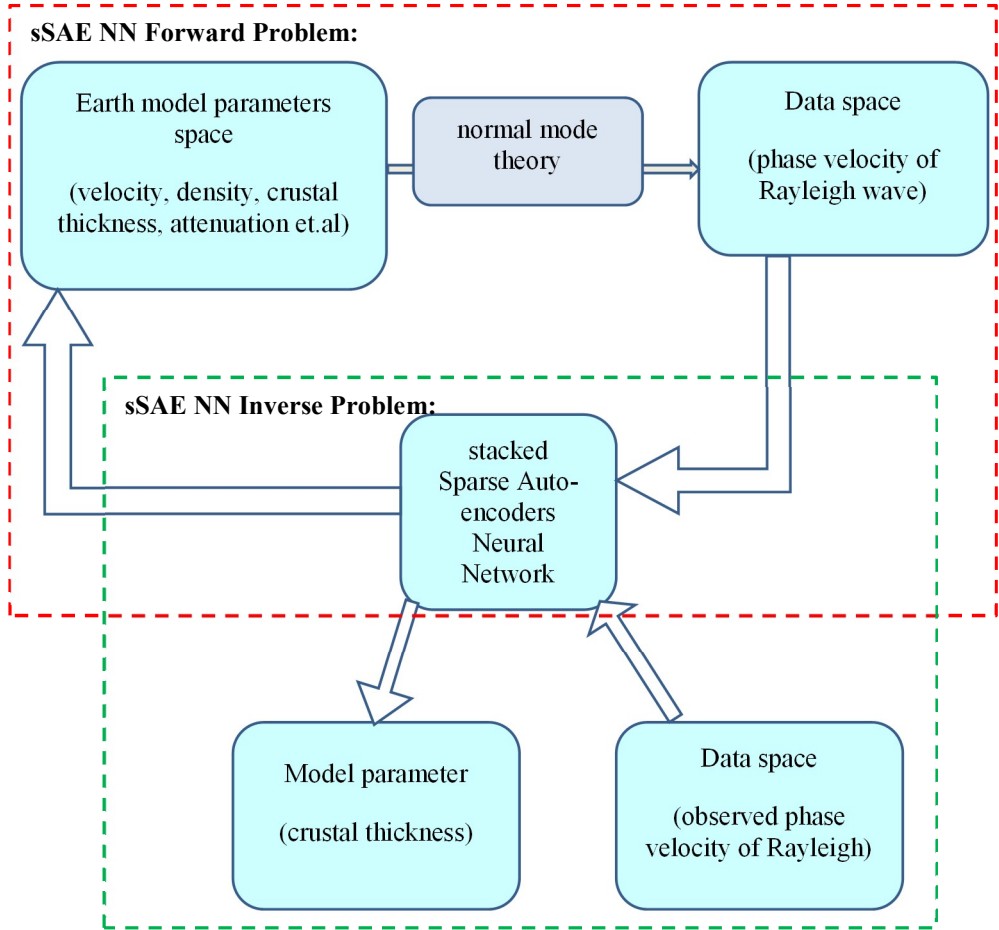

**Figure 3.** Crustal thickness inversion based on sSAE neural network composed of two parts:
Forward Problem and Inverse Problem.

3.1 data preparation

We closely follow the model parametrization methodology outlined in de Wit et al. (2014), which is based on the Preliminary Reference Earth Model(PREM, Dziewonski and Anderson,1981)and is parameterized on a discrete set of 185 grid points used by Mineos package(Masters et al., 2014). In addition, these models we have got show no correlations between physical parameters such as velocity, density, $\eta$ and attenuation profiles. As the model parametrization method mentioned above, we generate 500,000 synthetic models based on the 1-D reference models PREM, which are randomly drawn from the prior model distribution, also prior ranges for the various parameters in our model are given in tables A.2–A.4. of de Wit et al.(2014).We use the Mineos package to compute phase velocity for fundamental mode Rayleigh waves for all 500,000 synthetic 1-D earth models. As for observation data used in stage of inversion below, phase velocities are more sensitive to the deep structure than group velocity. Based on Rayleigh wave phase velocity from ambient noise(Xie et.al,2013) shown in Figure 4 averaged from 10 to 35mHz, we take these as input for our neural networks.

3.2 training sSAE deep learning neural network

**It is well known that the artificial neural network can approximate any nonlinear function to solve the nonlinear inverse problem by using a corresponding set of input-output pairs.These examples are presented to a network in a so-called training process, during which the free parameters of a network are modified to approximate the function of interests (de Wit et al. 2014). Here adopting sSAE deep learning neural network, detailed methods presented in section 2 above, we pre-train the neural network taking theoretical phase velocity of Rayleigh wave with random noise as inputs and theoretical crustal thickness as outputs to attain the initial weights and bias for neural network.**

**And then we take theoretical phase velocity of Rayleigh wave with random noise as input, and crustal thickness as output to fine-tune neural network as done in a traditional neural network.**

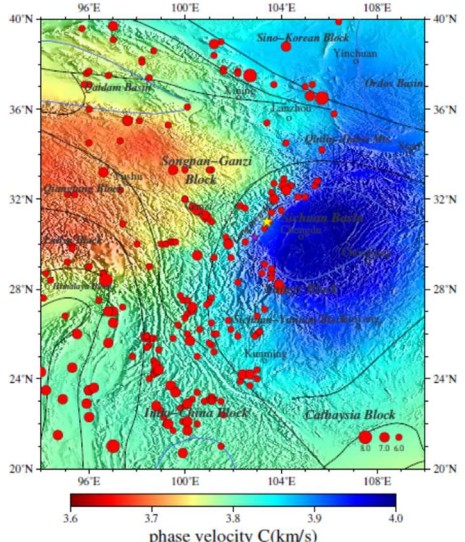

**Figure 4.** Average phase velocity of western Yangtze craton(Xie et al.,2013) from 10 to 35mHz.The black lines in the figure show structure lines. The blue lines show boundaries of sedimentary basins. The red dots show seismic events in this region from 1975 to 2015, and size of dot demonstrates size of magnitude from Ms 6.0 to Ms 8.0. The yellow and purple stars demonstrate Wenchuan and Lushan earthquakes respectively. These are same to Figure 4, Figure 6 and Figure 7.

**Figure 5.** The relationship between proportions of training data sets to test data sets and test errors.

How to find a satisfactory structure of neural network is a difficult problem because neural network training is sensitive to the random initialization of the network parameters. Therefore, as de Wit et al. (2014) pointed out that it is common practice to train several neural networks with different initializations, and subsequently choose the network which performs best on a given synthetic test data set, and the network which performed best on the test set is used to draw inferences from the observed data (de Wit et al. 2014). After trying many times, we find the proportion of training data set to test one is 3:1 is reasonable (Figure 5). The final test error depends not only on the number of input neurons, hidden layer and intermediate neurons, but also on the number of trainings, batch size and other optional parameters. In addition, the type of activation function, learning rate, zero masked fraction, and non-sparsity penalty value will affect final test errors. We give twelve cases and their corresponding test errors in table 1.

3.3 inverting crust thickness

Based on our all twelve neural networks, we invert Rayleigh phase velocities (10~35.0 mHz) to attain twelve crustal thickness models for eastern Tibet and western Yangtze craton. Considering not only the test errors of sSAE networks, but also misfits and correlation coefficients of our twelve models with crustal thickness models from other researches, we choose the network structure indicated by ※ in Table 1. We find the best fit crustal thickness model from sSAE (Figure 6).We compare our model with crustal thickness model from receiver function(Zhu et al.,2012),and the other two global crustal thickness models, CRUST2.0 from Bassin et al. (2000) based on refraction and reflection seismics as well as receiver function studies and the CUB2 model from Shapiro&Ritzwoller (2002)( Figure 7) who inverted a similar data set for crustal thickness using a Monte Carlo approach. The correlation coefficients and scatter plots of our model versus ZJS, our model versus CRUST2.0 and our model versus CUB2 (Figure 8) indicate that overall agreement between the three models. However, the agreements of our model with CUB2 and CRUST2.0 are better than with ZJS, since model ZJS attained from Zhu et.,al(2012) has

relatively sparse stations with poor data coverage and lower resolution. In addition, taking the Monte Carlo method (Hansen, 2013) using four processors only for 1000 iterations, it takes three weeks to invert the Xie (2013) data set to the crust thickness of the same region. As Shapiro, N. M,(2002) pointed out that the major disadvantage of this method is computational expense. Maybe the result is high resolution after many more iterations using Monte Carlo method. However, when we take use of sSAE, using only one processor, a six-layer network for 380,000 training samples, and 120,000 test samples for network training takes five hours, while a well-trained neural network inversion takes only a few seconds to complete.

Table1 deep learning neural network structures taking in this article

| sSAE Structure | parameters | | | | Error ×10⁻⁶ | CUB2 | | CRUST2.0 | | ZJS | |
|---|---|---|---|---|---|---|---|---|---|---|---|
| | Layers | D | E | F | $\times10^{-6}$ | G | H | G | H | G | H |
| [21 50 10 1] | Layer 1 | 0.3 | 10 | 1e4 | 262 | 7.29 | 0.79 | 7.68 | 0.80 | 9.12 | 0.71 |
| [21 50 10 1] | Layer 1 | 0.3 | 10 | 1e3 | 79.5 | 7.52 | 0.77 | 8.00 | 0.76 | 8.42 | 0.73 |
| [21 50 10 1] | Layer 1 | 0.3 | 10 | 1e2 | 27.83 | 7.29 | 0.78 | 7.32 | 0.79 | 7.98 | 0.72 |
| [21 50 10 1] | Layer 1 | 0.3 | 100 | 1e3 | 28.83 | 7.44 | 0.78 | 7.13 | 0.80 | 7.89 | 0.71 |
| [21 50 10 1] | Layer 1 | 0.3 | 100 | 1e2 | 11.29 | 7.34 | 0.79 | 6.61 | 0.82 | 7.79 | 0.68 |
| [21 50 10 1] | Layer 1 | 0.01 | 100 | 1e2 | 11.28 | 7.33 | 0.79 | 6.61 | 0.81 | 7.79 | 0.68 |
| [21 10 2 1] | Layer 1 | 0.01 | 100 | 1e2 | 15.73 | 7.08 | 0.79 | 6.67 | 0.82 | 7.91 | 0.68 |
| [21 100 50 20 1] | Layer 1 | 0.5 | 100 | 1e2 | 8.35 | 7.37 | 0.79 | 6.64 | 0.82 | 7.53 | 0.68 |
| [21 200 50 20 10 1] | Layer 1 | 0.5 | 100 | 1e2 | 7.62 | 7.32 | 0.79 | 6.69 | 0.81 | 7.59 | 0.68 |
| [21 200 100 50 20 10 5 1] ※ | Layer 1 | 0.5 | 100 | 1e2 | 7.22 | 6.75 | 0.80 | 6.70 | 0.82 | 8.00 | 0.69 |
| [21 200 100 50 20 10 5 1] | Layer 1 | 0.5 | 100 | 50 | 4.58 | 7.79 | 0.79 | 8.45 | 0.84 | 10.7 | 0.65 |
| [21 50 40 30 20 10 5 1] | Layer 1 | 0.5 | 100 | 50 | 6.04 | 7.62 | 0.78 | 8.35 | 0.83 | 10.3 | 0.66 |

In this article, we fixed the following three parameters in every situation: A-type of activation function(sigmoid); B-learning rate(1); C-zero masked fraction(0.5).

Various parameters: D-non-sparsity penalty, which is zero except for layer 1 in every sASE structure; E-number of epochs; F-size of batch.

G-RMS misfit of our result with other model; H-correlation coefficient of our result with other model.

※- selected sSAE neural network structure

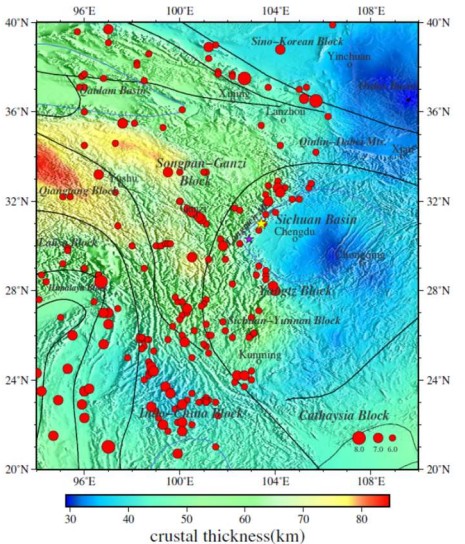

**Figure 6.** Crustal thickness of western Yangtze craton from this paper.

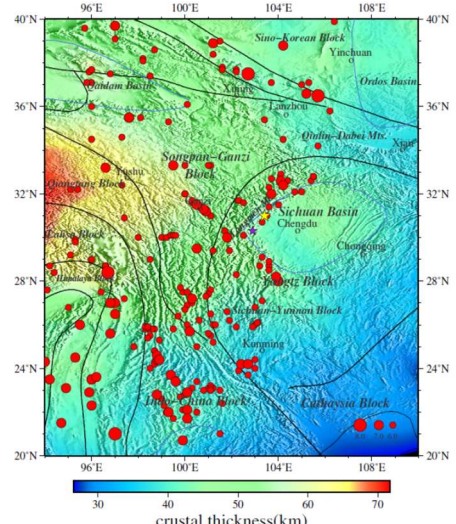

**Figure 7.** Crustal thickness of model CUB2 from Shapiro&Ritzwoller (2002). Note: Color scale is not exactly the same with figure 6.

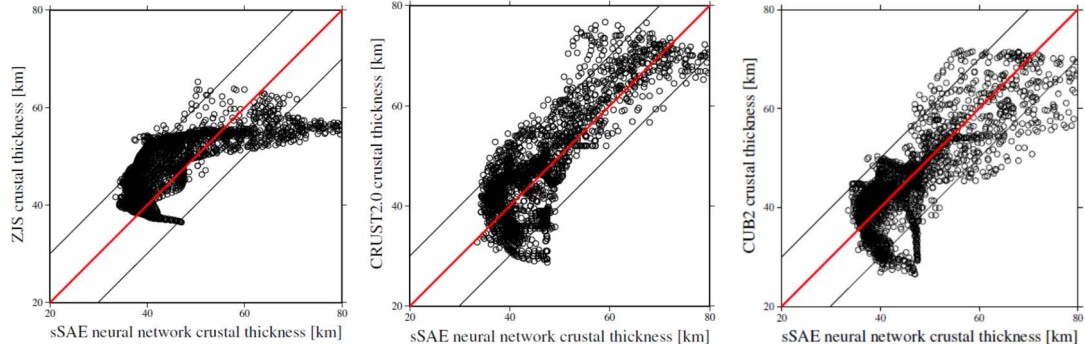

**Figure 8.**(From left to right) scatter plots of our model versus ZJS, our model versus CRUST2.0 and our model versus CUB2

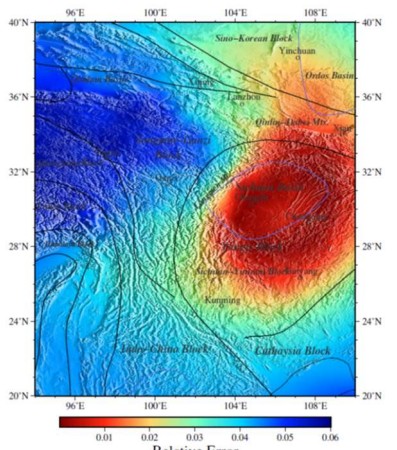

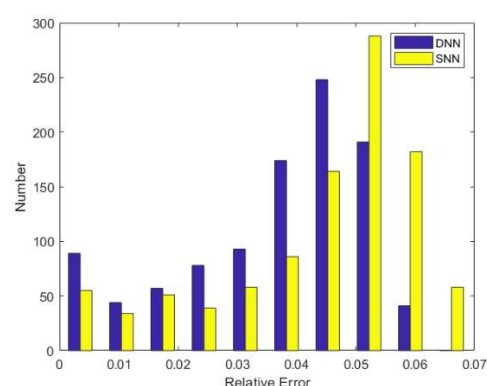

5    **Figure 9**. The relative error of the average phase velocity between the phase velocity calculated by our model and the phase velocity observed by Xie (2013).

**Figure 10**. The histogram of the relative error of the shallow neural network (SNN)and deep neural network(DNN).

## 4    Discussion

On the one hand, our results show that deep learning neural networks can effectively invert crustal
10    thickness because they have the ability to represent complex inverse functions :

A deep neural network can offer improvement over a shallow neural network as shown in Table 1.Test errors of deep learning neural network may be affected by the number of hidden layer in networks , The test error of deep learning neural network may be affected by the number of hidden layers in the network, which indicates that the more hidden layers, the smaller the test error. When the number of
15    hidden layer in networks adds from three to six, we can attain from Table 1, test error decreases from 2.6e-4 to 6.0e-6. In addition, the robustness of deep learning neural networks is strong. When the number of hidden layers in network reaches six, the change of the number of neurons in each layer has little influence on test errors, about 5.5e-6.

In addition, we conclude that different training parameters have different effects on training results.
20    We think that the size of batch is more important than epochs, as shown in Table 1.The size of batch decreases from 1e4 to 1e3 and test errors decrease from 2.6e-4 to 7.9e-5, however, epochs increase from

10 to 100, corresponding test errors change a little. The neural network structure indicated by ※ in table 1 reveals misfits of our model with model CUB2, CRUST2.0 and ZJS are relatively low with 6.75,6.70 and 8.0, and corresponding correlation coefficients are relatively high with 0.8, 0.82 and 0.69
25    respectively, however, test error is 7.22e-6 and is not minimum. This tells us that test error may not be the only criterion determining which neural network is best,because over-fitting may lead to small test errors.

Compared with works of Meier et al.(2007), in order to enhance robustness of neural networks, random noise is added to synthetic phase velocity as input in training progress. However, we have not

considered about the uncertainty of crustal thickness which should be revealed by deep mixture density network in a probabilistic manner in our future work.

On the other hand, we can attain the crustal thickness and resultant geodynamic implications in research region from our result. We find the relatively good agreement of our result (Figure 6) with CUB2(Fig.7),CRUST2.0 (Figure 8) and other recent researches(Qian, H. et al.,2018;Xu et al., 2016; Wang et al.,2015). All these models indicate that crustal thickness is deeper in the west of Longmen mountain than in the east of Longmen mountain. Which all showed an approximately 45-km-thick crust below the Sichuan basin, thickening beneath the Longmenshan and the high Tibetan plateau to about 60~80 km. Since we take use of regional high resolutional phase velocity model, our result reveals more details. In order to prove that these anomalies are persistent, only accidents, or inversion artifacts, we verify from two aspects: one is because our model gives a point estimate without uncertainty information, we propose the relative error between the phase velocity calculated based on our model and the phase velocity observed in the same region (Xie, 2013) to verify the uncertainty and non-uniqueness of the inversion results (Figure 9 and Figure 10). The relative error calculation formula is shown in Equation 4.

$$RE = \frac{|Pha_{cal}-Pha_{obs}|}{Pha_{obs}} \qquad (4)$$

RE- Realative Error of average phase velocity;
$Pha_{cal}$-calculated average phase velocity based on our model;
$Pha_{obs}$-observed average phase velocity from Xie(2013);

The relative error between East and West is significantly different(Figure 9). The reason is that in our research area, as shown in Figure 1b of Xie(2013), the stations in the east are much denser than in the west, and the Rayleigh wave measurements has higher resolution in the east. Therefore, compared with the west, the training data in the east is more dense, resulting in higher prediction accuracy of the neural network in the east and lower prediction accuracy in the west. Using the same parameters but for a traditional shallow neural network with three layers (one input layer, one intermediate layer and one output layer), we train this shallow neural network and obtain a relative error with the observed average phase velocity. The histogram error of the relative error of the shallow network is larger than that of our deep learning network (Figure 10). We can conclude that our model is more reliable in the eastern part of Longmen Mountain, especially in Chengdu, Qinling-Dabei fold belt, Xi'an and Ordos basins, and Sichuan-Yunnan block, so the crust thickness anomaly in these areas is worth explaining. Another reference to the results of other studies conducted by Wang (2010) in the same area, who attained the crustal thickness estimated by the H-k stacking method based on the broad band tele-seismic data recorded at 132 seismic stations in Longmen mountains and adjacent regions(26°~35°N,98°~109°E)(Figure11).

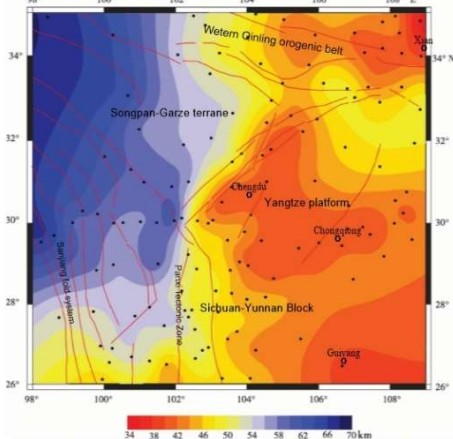

**Figure 11**. Contour map of crustal thickness estimated by h-k stacking analysis based on the receiver function(Wang ,2010, Small black solid circles indicate seismic station used by wang(2010)).

Our result reveals similar details with Wang(2010): the crustal thickness of the eastern Tibet plateau is complex and varies greatly. The average crust thickness is about above 60km, especially about 70-75km at Qiangtang block, under which there is a northward-dipping moho gradient zone. There is relatively shallow crust at Songpa-Ganzi block and is characteristic of decreasing in northwest-southeast orientation. Our model still shows some changes in the thickness of the crust in this area. For example, the thickness of the crust around Chengdu is relatively thin, especially in the northeastern part of Chengdu, about 50 km thick under the Qinlin-Dabei fold belt, and the thickness of the crust in the northeast to

Sichuan basin is about 45~48 km. In addition, crustal thickness around Xi'an and Ordos basin is about 35km. On the contrary, change of crustal thickness in Sichuan-Yunnan block varies greatly, with a thickness of about 60 km in the northwest and about 35 km in the southeast. From a geological viewpoint, the eastern Tibet and the western Yangtze craton has a very complex structure and tectonics, where several tectonic blocks, including the Yangtze Platform, the Songpan-Ganzi Fold System, the Qiangtang Block, and the Indochina Block, are interacting with each other. It is a site of important processes associated with the India-Asia collision and abutment against the stable Yangtze Platform, including strong compressional deformation with crust shortening and thickening, the plateau surface has been elevated to 4-5 km, and the Tibetan crust has doubled in thickness since the collision (Chen et al.,1996;Flesch et al., 2005;Wang,2010), east-west crustal extension, and strong earthquakes often occur on the active faults inside and on the edge of the plateau and are the most active seismic areas within the mainland. Based on the analysis of the distribution of the epicenters during 1970-2015, the results show that a large earthquake occurred in the brittle upper crust of the Longmenshan fault zone in Sichuan and Yunnan, and the crustal thickness changed sharply by about 10 km. Ms 8.0 Wenchuan earthquake in 2008 and Ms 7.0 Lushan earthquake in 2013 were caused by the reactions associated with the Songpan-Ganzi Fold System and the Qiangtang Block obliquely colliding with the Yangtze Platform. The reason may be that main fault cut off moho discontinuity where materials exchange between crust and mantle, and accumulated press triggers a series of earthquakes frequently.

## 5   Conclusion and remarks

Taking use of sSAE deep learning network, we present crustal thickness map of eastern Tibet and western Yangtze craton(Figure 7). The data sets consist of phase velocities of Rayleigh waves from Xie(2013) at discrete frequency of 10.0, 12.5, 15.0, 17.5, 20.0, 22.5, 25.0, 27.5, 30.0, 32.5,35.0mHz.We conclude that:

(1) Neural network structure is essential for inversion results, determined by the following parameters: the number of hidden layers, the number of neurons per layer,number of epoch, batch size, type of activation function, learning rate and non-sparsity penalty. We find that the number of parameters in hidden layers and the size of batch, are crucial for training neural networks. After many tests, the number of hidden layers was set to 6 ,the number of neurons was 200, 100, 50, 20, 10 and 5. The number of periods and the batch size were both set to 100, the activation function was sigmoid, and the learning rate was 1, higher resolution and more reliable crustal thickness model was attained.

(2) By training the deep network with six hidden layers and the traditional shallow network with only one hidden layer, the relative error between the phase velocity predicted by the deep network and the observed average phase velocity is smaller than that between the phase velocity predicted by the traditional shallow network and observed average phase velocity, indicating that the inversion result based on the deep network is more reliable than that of the traditional shallow network.

(3) Using only one processor, a six-layer sSAE network for 380,000 training samples, and 120,000 test samples for network training takes five hours, while a well-trained neural network inversion takes only a few seconds to complete. To complete the same inversion task, it takes three weeks to use the Monte Carlo method for four processors. We demonstrate that sSAE deep network inversion is more efficient than Monte Carlo inversion.

(4) Compared our model with current knowledge about crustal structure as represented by ZJS,CRUST2.0, CUB2,the overall agreement with these three models is very good, and agreement is generally better with CUB2 and CRUST2.0 which are attained from relatively dense stations with rich data coverage and higher resolution. Our result reveals more details in Chengdu, Qinling-Dabei fold belt, Xi'an and Ordos basins, and Sichuan-Yunnan block.

*Acknowledgements.* Our work was supported by the National Natural Science Foundation of China (Grant No. 41774095) the National Natural Science Foundation of China ( Grant No. 91755215).The authors are grateful to Xie for providing the phase velocity maps, N.M. Shapiro for making the model CUB2 available , Laske, G. for making the model CRUST1.0 available ,and Zhu for making the model ZJS available.

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
