# Peer review of "Inverting Rayleigh surface wave velocities for crustal thickness in eastern Tibet and the western Yangtze craton based on deep learning neural networks"

_Nonlinear Processes in Geophysics, 2018_

## Referee Comment (RC1) · H. Matchette-Downes (Referee) · 11 May 2018

**1   General comments**

To the best of my knowledge, the authors are the first to attempt to use deep neural networks (as opposed to shallow neural networks) to invert observations of surface wave dispersion for crust thickness (or any similar problem in seismology). However, major revisions are necessary to demonstrate that the method is working as intended,

and to show that it is an improvement over shallow neural networks (e.g. Meier et al., 2007), which are simpler. The manuscript also has some misleading statements and omissions. I suggest revisions below.

**2 Specific comments**

**2.1 Inclusion of noise in training data**

The authors do not mention whether or not the synthetic training data contain noise. A neural network trained on noise-free synthetic data will perform very poorly on real data containing noise (e.g. Meier et al., 2007, figure 8b). If noise was included in the training data, the authors should describe this. Otherwise, it should be included.

**2.2 Conversion from group velocity to phase velocity**

The authors calculate group velocity from a published phase velocity map using the standard formula (their equation 4). However, including both phase and group velocity will only add new information if the phase and group velocity are measured independently (as is commonly the case). Therefore it is misleading to include the calculated group velocity in this paper. The group velocity data should be removed from the study or replaced by group velocity data measured independently. (Generally phase velocity is more sensitive to deeper structure so it is easier to infer deep structure from phase velocity measurements.)

**2.3 Benefit of deep neural network versus shallow neural network**

A deep neural network is one with more than one hidden layer, whereas a shallow neural network has just one hidden layer. The additional complication of using a deep neural network is justified if the mapping has a hierarchical structure. For example, in image processing, it is common to move from the more elementary aspects of the input data (e.g. the values of the individual pixels) to intermediate parts (such as the distribution of edges) and finally to the most abstract aspects (such as the subject of the image). While it is undoubtedly true that the Earth has a hierarchical structure, ranging from individual grains to entire continents, the authors do not demonstrate that the dispersion data contain sufficiently complicated information to justify a deep neural network. The paper does not currently demonstrate that a deep neural network offers any improvement over a shallow neural network, such as that used by Meier et al. (2007). A comparison should be given.

**2.4 Non-unique solutions**

The authors focus on the non-linearity of the inverse problem, but they do not mention that it is also non-unique. Conventional optimisation of a neural network can lead to meaningless outputs for a non-unique mapping, as shown in figure 3b of Meier et al. (2007). Ideally, the method should be changed to solve for a probability distribution, for example using histogram or median networks (Devilee et al., 1999) or a mixture density network (Meier et al., 2007). Otherwise, the authors should attempt to quantify the range of non-uniqueness, or at least mention it in their discussion.

[Figure]

**2.5 Unattributed quotations**

Some explanatory sections are taken verbatim from other work, for example the paragraph beginning at 3.2:19 is identical to the second paragraph of section 3 of de Wit et al. (2014). These sections should be attributed, and either paraphrased or written in quotation marks.

**2.6 Meaning of 'data-driven'**

It is misleading to say that the method 'data-driven' (e.g. lines 1:9–11). The inversion is model-driven; it is trained using a large number of synthetic data which are generated using a known forward mapping (in this case, the calculation of dispersion by normal mode summation). The role of the neural network is to approximate the inverse relation apparent in the synthetic dataset. The description 'data-driven' is appropriate when the forward mapping is not known (or not used). An example would be speech processing, where the meaning of a word cannot be calculated from its audio waveform.

**2.7 Lateral resolution of crust thickness**

Figures 7 and 8 show a comparison of the crust thickness model in this study with the crust thickness model in Xie et al. (2013). Although the two models are based on the same data, the result in this study appears to resolve much finer features. The authors should explain how this higher resolution is achieved and whether it is justified.

**3 Corrections to the writing**

There are some errors in the writing, but I have not listed them in detail, in the expectation that the body of the text will change.

**4 References**

Devilee, Curtis & Roy-Chowdhury (1999), https://doi.org/10.1029/1999JB900273
Meier, Curtis & Trampert (2007), https://doi.org/10.1111/j.1365-246X.2007.03373.x
de Wit, Valentine & Trampert (2014), https://doi.org/10.1016/j.pepi.2014.09.004
Xie et al. (2014), https://doi.org/10.1002/jgrb.50296

---

## Author Comment (AC1) · 23 May 2018

We thank the referee for his working for this paper, who has given many good suggestions, which we are incorporated in this revised work. We answer all questions in attached file named "npg-2018-11_Author Reply(refree1).pdf". Also we upload the revised file named "npg-2018-11(revision).pdf" and. All these two files compress into a file named "npg-2018-11.zip".

Please also note the supplement to this comment:

[Figure]

https://www.nonlin-processes-geophys-discuss.net/npg-2018-11/npg-2018-11-AC1-supplement.zip

---

## Referee Comment (RC2) · Anonymous Referee #2 · 12 Jul 2018

General comments: This paper did the pioneer study to propose a deep learning neural networks method, called stacked sparse auto-encoder (sSAE), to obtain crustal thickness for eastern Tibet and western Yangtze craton. The input data are the phase and group velocities of Rayleigh waves. It is a good try to introduce a new methodology.

Major modifications in need

Questions need to be answered: 1) The paper has not told the reasons selected eastern Tibet and western Yangtze craton, while this study solves the problems. 2) What

is the theory of the sSAE to inverse the crustal thickness with phase and group velocities of Rayleigh waves? The details to get the dispersion data, phase velocities, and their combination for the sSAE inversion? 3) How to understand the inverted results for eastern Tibet and western Yangtze craton? The geological background needs to be added. 4) What are the merits of sSAE over other methods in fact? For instance, deep seismic sounding profile is the direct evidence of crustal thickness, what happens when two kinds of results are mapped together? Not the digital number listed in the table. 5) How to understand Table 1? 6) What is the difference between the results by sSAE and by other method? Not just the similarity.

---

## Author Comment (AC2) · 17 Jul 2018

We thank the referee for his working for this paper, who has given many good suggestions, which we are incorporated in this revised work. We answer all questions in attached file named "npg-2018-11_Author Reply(refree2).pdf". Also we upload the revised file named "npg-2018-11(revision2).pdf" and. All these two files compress into a file named "npg-2018-11(refree2).zip".

Please also note the supplement to this comment:

https://www.nonlin-processes-geophys-discuss.net/npg-2018-11/npg-2018-11-AC2-supplement.zip

---

## Author Response (AR1)

Response Letter

Author Reply

Authors Name:          Xian-QiongCheng , Qi-He Liu, Ping-Ping Li , Yuan Liu
Paper Name: Inverting Rayleigh surface wave velocities for crustal thickness in eastern Tibet and the western Yangtze craton based on deep learning neural networks

Revision Date:          8 /7/ 2018

Summary of Responses:

We thank the referees for their working for this paper, who have given many good suggestions, which we are incorporated in this revised work.

Below are the responses of work we have done.

**For refree1:**

| Comments and Suggestions | Response |
|---|---|
| **1. Inclusion of noise in training data**
The authors do not mention whether or not the synthetic training data contain noise. A neural network trained on noise-free synthetic data will perform very poorly on real data containing noise (e.g. Meier et al., 2007, figure 8b). If noise was included in the training data, the authors should describe this. Otherwise, it should be included. | We have trained our neural network on synthetic data with random noise and we have stated in the revision. |
| 2. **Conversion from group velocity to phase velocity**
The authors calculate group velocity from a published phase velocity map using the standard formula (their equation 4). However, including both phase and group velocity will only add new information if the phase and group velocity are measured independently (as is commonly the case). Therefore it is misleading to include the calculated group velocity in this paper. The group velocity data should be removed from the study or replaced by group velocity data measured independently. (Generally phase velocity is more sensitive to deeper structure so it is easier to infer deep structure from phase velocity measurements.) | We do not adopt the calculated group velocity and retrain our neural network on phase velocity only in the revision |
| 3. **Benefit of deep neural network versus shallow neural network**
A deep neural network is one with more than one hidden layer, whereas a shallow neural network has just one hidden layer. The additional complication of using a deep neural network is justified if the mapping has a hierarchical structure. For example, in image processing, it is common to move from the more elementary aspects of the input data (e.g. the values of the individual pixels) to intermediate parts (such as the distribution of edges) and finally to the most abstract aspects (such as the subject of the image). While it is undoubtedly true that the Earth has a | We retrain our neural network and find that more hidden layers can get more lower test errors than shallow neural network does , which can be demonstrated in table 1 |

| | |
|---|---|
| hierarchical structure, ranging from individual grains to entire continents, the authors do not demonstrate that the dispersion data contain sufficiently complicated information to justify a deep neural network. The paper does not currently demonstrate that a deep neural network offers any improvement over a shallow neural network, such as that used by Meier et al.(2007). A comparison should be given. | |
| 4. **Non-unique solutions**
 The authors focus on the non-linearity of the inverse problem, but they do not mention that it is also non-unique. Conventional optimisation of a neural network can lead to meaningless outputs for a non-unique mapping, as shown in figure 3b of Meier et al. (2007). Ideally, the method should be changed to solve for a probability distribution, for example using histogram or median networks (Devilee et al., 1999) or a mixture density network (Meier et al., 2007). Otherwise, the authors should attempt to quantify the range of non-uniqueness, or at least mention it in their discussion. | we have not considered about the uncertainty of crustal thickness which should be revealed by deep mixture density network in a probabilistic manner in our future work |
| 5.**Unattributed quotations**
 Some explanatory sections are taken verbatim from other work, for example the paragraph beginning at 3.2:19 is identical to the second paragraph of section 3 of de Wit et al. (2014). These sections should be attributed, and either paraphrased or written in quotation marks. | In the revision we re-write in quotation marks on identical to paragraph of Wit et al. (2014) |
| 6. **Meaning of 'data-driven'**
 It is misleading to say that the method 'data-driven' (e.g. lines 1:9–11). The inversion is model-driven; it is trained using a large number of synthetic data which are generated using a known forward mapping (in this case, the calculation of dispersion by normal mode summation). The role of the neural network is to approximate the inverse relation apparent in the synthetic dataset. The description 'data-driven' is appropriate when the forward mapping is not known (or not used). An example would be speech processing, where the meaning of a word cannot be calculated from its audio waveform. | Our manuscript aims at inverse problem, so meaning of data-driven in the manuscript is that we have no idea of inverse relationship , although the forward mapping is known. That is, we have no model describing how to infer crustal thickness from phase velocity. So we think this belongs to data-driven problem. |
| 7. **Lateral resolution of crust thickness**
 Figures 7 and 8 show a comparison of the crust thickness model in this study with the crust thickness model in Xie et al. (2013). Although the two models are based on the same data, the result in this study appears to resolve much finer features. The authors should explain how this higher resolution is achieved and whether it is justified. | In the discussion we talk out our result resolve much finer features than other models, and these finer features revealed by our result is consistence with Wang et.al(2010) who attained the crustal thickness estimated by the H-k stacking method based on the broad band tele-seismic data. We think this higher resolution is achieved as deep sSAE works very well in learning useful high-level feature for better representation of input raw data. |
| 8. **Corrections to the writing**
 There are some errors in the writing, but I have not listed them in detail, in the expectation that the body of the text will change | We check the English sentence by sentence and upload revised manuscript |

Response Letter

**For refree2:**

| Comments and Suggestions | Response |
|---|---|
| 1) **The paper has not told the reasons selected eastern Tibet and western Yangtze craton, while this study solves the problems.** | We add the reason selected eastern Tibet and western Yangtze craton in revised paper in page 1 from line 29 to line 36 |
| 2) **What is the theory of the sSAE to inverse the crustal thickness with phase and group velocities of Rayleigh waves? The details to get the dispersion data, phase velocities, and their combination for the sSAE inversion?** | (1).the theory of the sSAE to inverse the crustal thickness with phase and group velocities of Rayleigh waves is finding the relationship between the two variables by machine learning. A stacked autoencoder is a neural network consisting of multiple layers of sparse autoencoders in which the outputs of each layer is wired to the inputs of the successive layer. Firstly taking therotical phase velocities with random noise as inputs and theoretical crustal thickness as outputs we train the deep learning neural networks. Then taking observed phase velocities as input into the trained neural network and can attain an output. The output is seen as estimation of real crustal thickness. (2).Theoritical dispersion based on normal mode [Dziewonski,1981]. $c = {}^{n\omega_l\alpha}\!\big/\!\left(l+\frac{1}{2}\right)$ (in which $c$-phase veloicty; n-radial order;$l$-angular order; $\omega$ - eigenfrequency; $\alpha$ -radius of earth.) Observed dispersion is based on (Xie et.al,2013)from ambient noise, which Rayleigh wave phase speed measurements are obtained from cross correlations of vertical-component ambient noise,the vertical-vertical (Z-Z) cross correlations. |
| 3) **How to understand the inverted results for eastern Tibet and western Yangtze craton? The geological background needs to be added.** | The geological background about eastern Tibet and western Yangtze craton are added in revised paper in page 10 from line 23 to line 36 |
| 4) **What are the merits of sSAE over other methods in fact? For instance, deep seismic sounding profile is the direct evidence of crustal thickness, what happens when two kinds of results are mapped together? Not the digital number listed in the table.** | Comparable crustal thickness model crust2.0 adopted in this paper based on refraction and reflection seismics as well as receiver function studies, the comapration result is shown in the middle of figure 8 |
| 5) **How to understand Table 1?** | Table 1 states that different test errors and comparison results between our model and other models based on 11 different neural networks. |
| 6) **What is the difference between the results by sSAE and by other method? Not just the similarity.** | Compared with other crustal thickness models,our result reveals more details discussed in paper in page 10 from line 9 to line 20. And we add the diffference in tha abstact from line 19 to line 24 |
| | |

[revised manuscript text omitted]

---

## Referee Report (RR1)

Dear editor,

Dear authors,

This manuscript entitled "***Inverting Rayleigh surface wave velocities for crustal thickness in eastern Tibet and the western Yangtze craton based on deep learning neural networks***" presents an example of applying the deep learning neural network method to invert for crustal thickness using surface waves.

While I found the method that the authors introduced is quite interesting, I do have some serious concerns about the readability of the manuscript. Also, it is not clear to me whether the authors' approach is indeed superior to conventional nonlinear inversion techniques (e.g. the Bayesian Monte Carlo algorithm used in Shapiro & Ritzwoller, 2002).

**In summary, I would like to recommend that this manuscript to be published with major revision.**

Please find more detailed comments and suggestions below.

**Grammar mistakes**

I must say, unfortunately, that there are grammar mistakes almost everywhere in the manuscript. Below I list a few of them:

1) P1L15-L20: "… deep learning neural network based on data driven with the highly nonlinear mapping ability **can be widely used by geophysical inversion method**"

This sentence simply does not make sense to me. I suggest changing it to "**widely used by geophysicists**" or "**widely used by researchers**"

2) P1L35: "Especially, **a newest** crust model called crust1.0…"

I've never seen such a saying of "a newest", I think what the authors mean is "***one of the newest models***"

3) P2L7: "their results are heavily depended on…"

 It should be "their results heavily depend on…"

4) P3L5: "The more deep…"

The deeper

5) P4L8: "without sparsity **constrain**"

constraint

6) P6L8: "according (4)"

according **to** (4)

7) P6L12: "As we all know,…"

It is NOT appropriate to use this phrase in scientific writing.

8) P9L2: "is **consistence** with"

consistent

I understand that it is common to make grammar mistakes in a manuscript. However, there are so many mistakes in this manuscript making it hard to read and understand. Please try to rewrite the sentences making them more readable.

**Significance of the manuscript**

The authors used the deep learning neural network algorithm to invert for crustal thickness and compared their results with the crustal model presented in Shapiro & Ritzwoller (2002). It seems to me that the manuscript is not significant enough for publication because the authors failed to demonstrate two important things:

**1. Is the new method indeed better?**

I do not see any advantages of the authors' method compared with the Monte Carlo method used in Shapiro & Ritzwoller (2002). Indeed, as the authors mentioned in their manuscript, their result reveals more details (P8L15). However, this could be because the phase/group speed maps the authors used in their inversion have better resolution. It is entirely possible that Shapiro & Ritzwoller (2002)'s method could also yield crustal thickness map with more details by applying to Xie et al.(2013)'s datasets. In short, **the author need to demonstrate that their model has higher resolution because the method they used is different, not because of the differences in the datasets.**

**2. Is the authors' model indeed better than Shapiro & Ritzwoller (2002)?**

The authors' model reveal more details, but **details do NOT necessarily mean better**. It is possible that the small-scale features in the authors' model are artefacts and thus unreal.

In P9L2, the authors argued that the detailed information of their model is consistent with Wang et al.(2010), however, they failed to provide more details for the readers. The readers will have no idea how similar the authors' model is compared with Wang et al.(2010), because Wang's paper was written in Chinese. I strongly suggest the authors provide a figure comparing their model with Wang et al.(2010)'s model to demonstrate that their model indeed **reveals more details that are captured in another study**.

---

## Referee Report (RR2)

Whereas Xie et al., 2013 used their phase-velocity measurements from 10 to 65 seconds to invert for shear-velocities for depths between 10 and 85 km, Cheng et al. only used velocities from 10 to 35 seconds to invert for Moho depth.

In this region, as shown in their results, the Moho depth very from 30 km to 75 km.

Do Rayleigh-waves sampled up to 35s have enough sensitivities at depth of 75 km?

Maybe the author can provide some sensitivity kernels to convince the readers.
* * *
Eastern Tibet is highlighted by very slow velocity anomalies, whereas very fast velocities are found beneath the Sichuan Basin (Figure 4).

However, in Figure 6, a thick crust is found beneath eastern Tibet, but the crustal thickness beneath the Sichuan basin seems similar to its surrounding.

Recent papers on Sichuan basin show a relatively thick crust beneath the Sichuan basin (Shapiro & Ritzwoller, 2002; Wang et al., 2003; Xie et al. 2013; Legendre et al., 2014).

Legendre, C. P., F. Deschamps, L. Zhao, S. Lebedev, and Q.F. Chen (2014), Anisotropic Rayleigh wave phase velocity maps of eastern China, J. Geophys. Res. Solid Earth, 119, 4802-4820, doi:10.1002/2013JB010781.

Wang, C., Wu, J., Lou, H. et al. Sci. China Ser. D-Earth Sci. (2003) 46(Suppl 2): 254. https://doi.org/10.1360/03dz0020

In addition, I would suggest to the authors to expand a bit their bibliographic record for this region.

Many papers have been published in recent years but only few have been properly cited.
* * *
In some part of the manuscript, some rephrasing seems necessary:
- page 1, line 34 "the nature of the two blocks is different, especially the special topography". What is the special topography?
- page 2, lines 10-13.
- page 2, line 47: stacked Sparse Auto-encoders (sSAE) --> stacked Sparse Auto-Encoders (sSAE).
- page 9, caption of figure 10: estimted --> estimated
- page 10, line 32: xie --> Xie
* * *
References:
There are many problems with the references, a few are listed below and need to check carefully.
In addition, several papers are in Chinese, and would not be helpful to the international community. Please find some references in English in addition whenever it is possible.

Liu et al.(2015) - page 3, line 54
--> doesn't appear in the references
Gao X., Wang W.M, Yao, Z. X.: Crustal structure of China mainland and its adjacent regions. Chinese J .Geophys, (in Chinese) , 48,3,591-601,2005.
--> doesn't appear in the text
Huang J. P., Fu R. S. Xu P., et al.: Inversion of gravity and topography data for the crust thickness of China and its adjacency, Acta Seismologica Sinica(in Chinese),28,3,250-258,2006.

--> doesn't appear in the text
Huang et al., 2008
--> doesn't appear in the references
Lampinen, J. &Vehtari, A.: Bayesian approach for neural networks -review and case studies, Neural Networks, 14(3), 257–274,2001.
--> doesn't appear in the text

---

## Referee Report (RR3)

**Second review of 'Inverting Rayleigh surface wave velocities for crustal thickness in eastern Tibet and the western Yangtze craton based on deep learning neural networks'**

**Reviewer:** Harry Matchette-Downes

**Date:** 24th January 2019

**Journal:** Nonlinear Processes in Geophysics

The revised manuscript addresses some of the issues raised during the review process: inclusion of noise in training data, conversion from group velocity to phase velocity, unattributed quotations, and quality of writing. However, the new manuscript still has some flaws which should be addressed before this work is published.

Firstly, the manuscript does not mention non-uniqueness, which is a crucial part of this inverse problem. The authors should at least discuss the possible errors introduced by neglecting non-uniqueness.

The most serious issue regards the key proposition of the paper: that deep neural networks can perform a better inversion than similar, but simpler, existing methods using shallow neural networks. This point is not convincingly

shown. For example, the if we overlay the comparison between the new model and the CUB2 model (figure 8, right panel) with the same figure from Meier et al. (2007; figure 16, middle panel), there is no apparent difference between the two approaches (see figure below). In addition, figure 9 shows misfits of 2 to 9%, which are large compared to the data uncertainties reported by Xie et al. (2013, figure 4), which are 0.5 to 2%. To demonstrate that the new method is an improvement, the authors should present similar plots for (1) an optimised shallow neural network (2) their optimised deep neural network. These should be accompanied by histograms of the fractional error in crust thickness, and examples of dispersion curves showing the model fit.

[Figure]

Lastly, the systematic east-west variation in misfit is not explained. In figures 6 and 7, we see a comparison with the model of Shapiro and Ritzwoller (2002), and it seems that the crust is much thicker in Tibet for the new model, which may explain the large errors. This is typical for crust thickness inversion using

group velocity in Tibet (see Meier et al., 2007, middle row of first panel of figure 15). The authors should inspect the dispersion curve fits and try to explain the source of the misfit.

**1 References**

Meier, Curtis & Trampert (2007), https://doi.org/10.1111/j.1365-246X.2007.03373.x

Shapiro & Riztwoller (2002), https://doi.org/10.1046/j.1365-246X.2002.01742.x

Xie et al. (2013), https://doi.org/10.1002/jgrb.50296

---

## Author Response (AR2)

Author Reply

Authors Name:       Xian-QiongCheng , Qi-He Liu, Ping-Ping Li , Yuan Liu
Paper Name: Inverting Rayleigh surface wave velocities for crustal thickness in eastern Tibet and the western Yangtze craton based on deep learning neural networks

Revision Date:       12 /9/ 2018

Summary of Responses:

We thank the referee for his working for this paper, who has given many good suggestions, which we are incorporated in this revised work.

Below are the responses of work we have done.

For refree1:

| Comments and Suggestions | Response |
|---|---|
| 1) **Grammar mistakes** | Thanks to the reviewer for the suggestions. We check the English sentence by sentence and upload revised manuscript. |
| 2) **Is the new method indeed better?** | In order to demonstrate if these anomalies are persistent, are mere accidents, or are artifacts of the inversion, we refer to the result of research in the same region from Wang(2010), who attained the crustal thickness estimated by the H-k stacking method based on the broad band tele-seismic data recorded at 132 seismic stations in Longmen mountains and adjacent regions(26°~35°N,98°~109°E)(Figure 9 in the article). Our result reveals similar details with Wang(2010) and indicates these anomalies are persistent. |
| 3) **Is the authors' model indeed better than Shapiro & Ritzwoller (2002)?** | Taking the Monte Carlo method (Hansen, 2013) and using four processors for only 1000 iterations, it takes three weeks to invert the Xie (2013) data set to the crust thickness of the same region , and the result shown below indicate that overall agreement between our and this result. Although this result shows singular values in some places , maybe the result is high resolution after many more iterations using Monte Carlo method. However, in our approach, our training process took less than 6 hours and the inversion process took a few minutes. Compared to our method, Monte Carlo method is computational expense. |

[Figure]

For refree2:

| Comments and Suggestions | Response |
| --- | --- |
| A significant result of this manuscript would be if the sSAE neural network achieved better results than a simple 'shallow' neural network (e.g. Meier et al., 2007). However, achieving lower test errors is not sufficient to show that the complicated neural network is better than the simple one. More work is needed to demonstrate that the sSAE is not over-fitting the training data. For example, it is necessary to state the total number of parameters in the neural network compared to the number of training data points. If the total number of parameters is large compared to the number of training data, then further work is necessary to check for over-fitting, such as regularisation. | Thanks to the reviewer for the suggestions. We try our best to avoid overfitting based on neural network development. Our model uses 380,000 data as the training set and 120,000 data as the test set. The two data are separated and the iteration is stopped when the error of the test set is not falling. Therefore, the early stop mechanism we used in this article to avoid over-fitting problems. In addition, in the fine tuning of the model, a second-order norm regularization method is also used to avoid overfitting. On the other hand, the number of our model weight parameter is 30455, the number of bias parameters is 386, a total of 30841 parameters, and the number of parameters is much smaller than the number of training data. |

---

## Author Response (AR5)

**Author Reply**

Authors Name:       Xian-QiongCheng , Qi-He Liu, Ping-Ping Li , Yuan Liu
Paper Name: Inverting Rayleigh surface wave velocities for crustal thickness in eastern Tibet and the western Yangtze craton based on deep learning neural networks

Revision Date:       3 /19/ 2019

Summary of Responses:

We thank Dr Richard Gloaguen for his workings for this paper, who have given many good suggestions, which we are incorporated in this revised work.

Below are the responses of work we have done.

For Dr Harry Matchette-Downes:

| Comments and Suggestions | Response |
|---|---|
| 1) 1- the PDF I received had very bad quality. Figures were all over the place and I could not see labels. | This time I have uploaded the pdf file and corresponding word file. |
| 2- the conclusions need to be rewritten | The conclusions have been to be rewritten in the revised manuscript |
| 2.a (1) a belief is not scientific. reformulate | We conclude deep learing outperform traditional shallow neural network and Monte Carlo inversion in rewritten conclusions (2) and (3) |
| 2.b (1) state how you determine the parameters accurately | In the rewritten conclusion (1), key parameters for neural network are given. |
| 2.c (2) "different networks produced different results" Those statements are not acceptable. Be precise. | This statement is replaced by precise value of key parameters for neural network in rewritten conclusion (1) |
| 2.d (2) there are ways to estimate overfitting. | Overfitting is a relatively complex problem of neural network inversion, and we have not discussed this issue in the rewritten conclusions. |
| 2.e (3) what is the added value of your model. Does it contradict previous models? | In conclusion (4) we conclude that our model has the overall agreement with previous models and our result reveals more details somewhere |
| 2.f (4) "Geophysical inversion is very complex, so it is necessary to analyze and enhance neural network to adapt to these complicated problems." This is not a conclusion. It's a biased statement | In the rewritten conclusions we delete this statement |